# Prismer: A Vision-Language Model with Multi-Task Experts

**Shikun Liu**[1,2*]    **Linxi Fan**[2]    **Edward Johns**[1]    **Zhiding Yu**[2]
**Chaowei Xiao**[2,3]    **Anima Anandkumar**[2,4]
[1]*Imperial College London*    [2]*NVIDIA*    [3]*University of Wisconsin, Madison*    [4]*Caltech*

**Reviewed on OpenReview:** *https://openreview.net/forum?id=R7H43YD6Lo*

## Abstract

Recent vision-language models have shown impressive multi-modal generation capabilities. However, typically they require training huge models on massive datasets. As a more scalable alternative, we introduce Prismer, a data- and parameter-efficient vision-language model that leverages an ensemble of task-specific experts. Prismer only requires training of a small number of components, with the majority of network weights inherited from multiple readily-available, pre-trained experts, and kept frozen during training. By leveraging experts from a wide range of domains, we show Prismer can efficiently pool this expert knowledge and adapt it to various vision-language reasoning tasks. In our experiments, we show that Prismer achieves fine-tuned and few-shot learning performance which is competitive with current state-of-the-arts, whilst requiring up to two orders of magnitude less training data. Code is available at `https://github.com/NVlabs/prismer`.

## 1 Introduction

Large pre-trained models have demonstrated exceptional generalisation capabilities across a wide range of tasks. However, these capabilities come at a hefty cost in terms of computational resources required for training and inference, as well as the need for large amounts of training data. In the language domain, models with hundreds of billions of learnable parameters typically require a compute budget on the yottaFLOP scale (Chowdhery et al., 2022; Brown et al., 2020; Black et al., 2022; Rae et al., 2021).

The problems in vision-language learning are arguably more challenging. This domain is a strict super-set of language processing, whilst also requiring extra skills unique to visual and multi-modal reasoning. For example, many image captioning and visual question answering problems require the model to be capable of fine-grained object recognition, detection, counting, and 3D perception (Antol et al., 2015; Chen et al., 2015). A typical solution is to use a massive amount of image-text data to train one giant, monolithic model that learns to develop these task-specific skills from scratch, simultaneously, and within the same generic architecture.

Instead, we investigate an alternative approach to learning these skills and domain knowledge via *distinct and separate sub-networks*, referred to as "experts". As such, each expert can be optimised independently for a specific task, allowing for the use of domain-specific data and architectures that would not be feasible with a single large network. This leads to improved training efficiency, as the model can focus on *integrating* specialised skills and domain knowledge, rather than trying to learn everything at once, making it an effective way to *scale down* multi-modal learning.

To achieve this, we propose Prismer[1], a visually conditioned autoregressive text generation model, trained to *better use diverse pre-trained task experts* for open-ended vision-language reasoning tasks. Prismer's key design elements include i) powerful vision-only and language-only models for *web-scale knowledge* to construct our core network backbones, and ii) multi-task vision experts encoding multiple types of visual information,

---

*Corresponding Author: shikun.liu17@imperial.ac.uk. Work done during an internship at NVIDIA.

[1]The model name "Prismer" draws from the analogy to an optical prism which breaks a white light into a spectrum of colours, and here we break down a single reasoning task into diverse domain-specific reasoning.

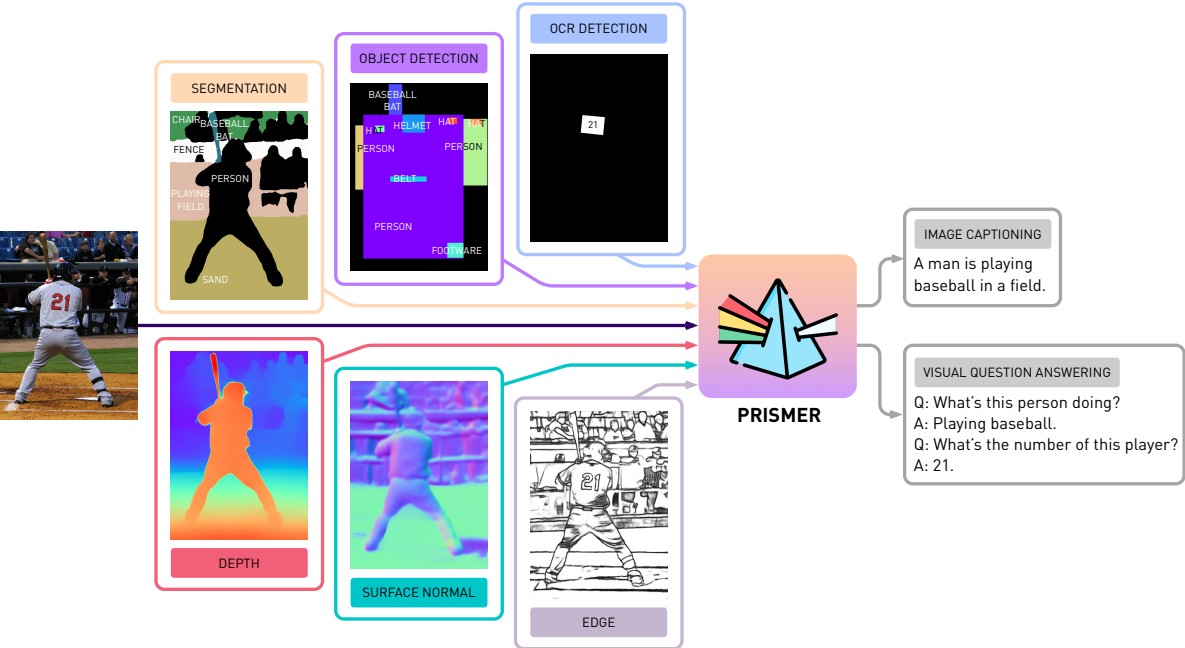

Figure 1: **Prismer model overview.** Prismer is a data-efficient vision-language model that leverages diverse pre-trained experts through its predicted multi-task signals. It can perform vision-language reasoning tasks such as image captioning and visual question answering. The analogy is with an optical prism: Prismer splits a single reasoning task into diverse domain-specific reasoning.

including *low-level vision signals* such as depth, and *high-level vision signals* such as instance and semantic labels, as a form of *auxiliary knowledge*, directly from their corresponding network outputs. All expert models are individually pre-trained and frozen, and are connected through some lightweight trainable components which contribute to roughly 20% of the total network parameters.

Despite Prismer being trained on only 13M publicly available image/alt-text data examples, it shows strong multi-modal reasoning performance in tasks such as image captioning, image classification, and visual question answering, competitive with many state-of-the-art vision-language models (Alayrac et al., 2022; Wang et al., 2022a; 2021), that were trained with one or two orders of magnitude more data. Finally, we conduct an in-depth analysis of Prismer's learning behaviours and observe some encouraging properties. For example, i) Prismer exhibits *strong robustness against the inclusion of noisy experts*, and ii) the learning performance also scales favourably with increases in both the *quantity* or *quality* of experts.

## 2  Related Work

**Vision-Language Models (VLMs)**   Inspired by the breakthrough of transformers in the language domain (Vaswani et al., 2017; Devlin et al., 2019), early works aimed to model the vision-language relationship using a shared network based on transformers in a *single-stream* design (Li et al., 2020a; Chen et al., 2020; Li et al., 2020b; Su et al., 2020). These works usually leverage a pre-trained object detector, encoding images as sequences of *visual words*, parameterised by object- or region-level features. Prismer takes a slightly different approach by using pre-trained models to provide their output predictions as auxiliary signals, whilst still relying on the original images to encode visual features.

Another line of works encodes vision and language features in separate networks in a *dual-stream* design, where the vision-only and language-only features are aligned through contrastive learning (Radford et al., 2021; Zhai et al., 2022; Jia et al., 2021; Li et al., 2021). These works typically focus on close-ended multi-modal alignment tasks such as image-text classification and retrieval. In contrast, Prismer's vision encoder also

aligns its vision features with the language embedding through pre-training with contrastive learning, but with a greater emphasis on multi-modal generation tasks.

Both single- and dual-steam VLMs in the past years have often been pre-trained with a combination of multiple objectives, such as masked language modelling, masked region modelling, word-region alignment, visual grounding and more (Li et al., 2020a; Cho et al., 2021; Li et al., 2022; 2021; Lu et al., 2019). These multiple objectives can make the training process more complex and require careful balancing of the different losses. Prismer adopts a different approach, aligning with recent developments in VLMs that focus on language generation, and only require a single autoregressive training objective (Wang et al., 2022a; 2021; Hu et al., 2022). Despite the reduced complexity, training these large-scale VLMs is data intensive and computationally demanding, often requiring billions of training data. To overcome these challenges, Prismer leverages powerful pre-trained task-specific expert models for data-efficient training. Unlike another set of works that prioritise in-context capability by conditioning on a large frozen language model with no task-specific fine-tuning (Eichenberg et al., 2021; Tsimpoukelli et al., 2021; Alayrac et al., 2022), Prismer focuses on fine-tuned performance with an emphasis on parameter efficiency, using smaller but diverse pre-trained experts.

**Multi-task and Auxiliary Learning**  Multi-task learning and auxiliary learning aim to train models to predict multiple outputs (such as semantic segmentation, object detection, and depth estimation) from a single input, thereby improving the performance across one or multiple tasks. This is often achieved through the design of effective multi-task networks that balance task-shared and task-specific features (Liu et al., 2019b; Misra et al., 2016; Sun et al., 2020; Xu et al., 2018), or through the explicit modelling of task relationships (Liu et al., 2019a; 2022; Navon et al., 2021; Zamir et al., 2018; Fifty et al., 2021). Recently, multi-task learning has been further generalised to unify vision-only, language-only, and vision-language tasks by considering them within a sequence-to-sequence framework (Wang et al., 2022b; Lu et al., 2022; Zhu et al., 2022). Prismer also employs multiple tasks, specifically in the vision domain, similar to these methods, but uniquely uses them solely as input, serving as auxiliary knowledge. Prismer is more related to works such as (Bachmann et al., 2022; Ghiasi et al., 2021), which utilise pre-trained experts to create pseudo labels for multi-task self-training. However, whilst those methods focus on learning task-agnostic features through multi-task supervision, Prismer focuses purely on multi-modal reasoning with a single-task objective.

**Unifying Pre-trained Experts**  The utilisation of diverse pre-trained domain experts for multi-modal reasoning has been investigated in previous studies. Socratic models (Zeng et al., 2022) use language as a one-way communication interface to connect different pre-trained experts. ViperGPT (Surís et al., 2023) and Visual Programming (Gupta & Kembhavi, 2023) harness the in-context learning capabilities of large language models, breaking down complex multi-modal reasoning into modular programs, which are then solved sequentially by leveraging pre-trained vision experts through APIs. The aforementioned methods excel at modular problem decomposition and establishing connections among pre-trained experts, and thereby being limited to zero-shot multi-modal reasoning within the domains on which the experts were pre-trained, and errors predicted by previous experts can be carried forward to future experts. However, Prismer stands out with a distinct objective by aiming to better bridge these pre-trained experts through a unified architecture design. As such, Prismer aims to create a more seamless collaboration between these experts, ultimately optimising multi-modal reasoning in a more integrated manner, and more robust to non-optimal experts.

Finally, we would like to highlight the distinction between the concept of "experts" defined in "Mixture of Experts (MoE)"(Riquelme et al., 2021; Nguyen & Chamroukhi, 2018; Masoudnia & Ebrahimpour, 2014) and in Prismer. In MoE, the "experts" are sub-modules in a single network, interconnected through their corresponding gating networks, encoding *implicit knowledge* guided by a shared training objective. On the other hand, in Prismer, the "experts" are independently pre-trained models, encoding *explicit knowledge* based on their pre-trained tasks or domains.

## 3   Prismer: Open-ended Reasoning with Multi-Task Knowledge

In this section, we introduce the Prismer model, a type of vision-language generative model that takes multi-task signals as input, and outputs free-form text.

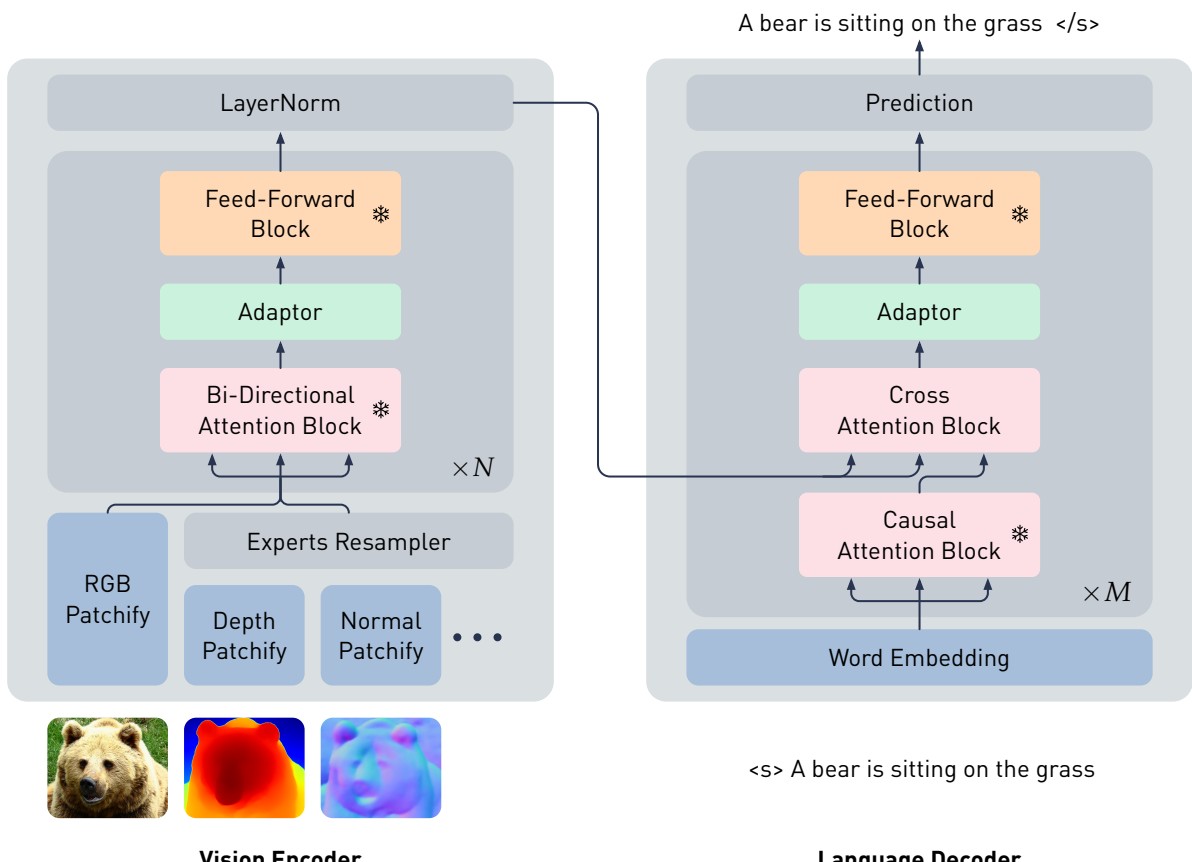

Figure 2: **Prismer architecture design overview.** Prismer has two main trainable components: the Experts Resampler that converts variable multi-task signals to a fixed number of outputs, and the Adaptor that enhances the model's expressivity for vision-language reasoning. To ensure that the model takes advantage of the rich domain-specific knowledge encoded in the pre-trained experts, the majority of network weights are frozen during training, as represented by ❄.

## 3.1   Model Overview

The design of the Prismer model is illustrated in Fig. 2. Prismer is an encoder-decoder transformer model (Vaswani et al., 2017) that leverages a library of existing pre-trained experts. It consists of a vision encoder and an auto-regressive language decoder. The vision encoder takes an RGB image and its multi-task labels as input (*e.g.* depth, surface normal, segmentation labels, predicted from the frozen pre-trained experts), and outputs a sequence of RGB and multi-task features. The language decoder is then conditioned on these multi-task features via cross attention, and produces a sequence of text tokens.

Prismer is designed to leverage pre-trained experts whilst keeping the number of trainable parameters to a minimum. To do this, the network weights of the pre-trained experts are frozen to maintain the *integrity of their learned knowledge* and prevent catastrophic forgetting (Kemker et al., 2018; Kirkpatrick et al., 2017). To link the multi-task knowledge as well as the vision and language parts of Prismer, we insert two parameter-efficient trainable components: *Experts Resampler* and *Adaptor*. The Experts Resampler is used in the vision encoder to map a variable length of multi-task signals to a sequence of multi-task features with a *fixed length*. The Adaptors are inserted in each transformer layer of the vision and language parts of the model to better adapt the pre-trained experts to new tasks and modalities.

Prismer is a *generative* model, and we re-formulate all vision-language reasoning tasks as a *language modelling or prefix language modelling* problem. For example, given the input image along with its multi-task tokens (predicted with the multi-task experts) and a question as the prefix, the model generates the answer for the

visual question answering task; given the input image along with its multi-task tokens, the model generates its caption for the image captioning task. Once we have a prefix prompt, we may either sample the output text in an autoregressive manner, as in an *open-ended* setting; or we may rank the log-likelihood from a fixed set of completions, as in a *closed-ended* setting.

## 3.2 Pre-trained Experts

In Prismer, we include two types of pre-trained experts:

**Backbone Experts**  The vision-only and language-only pre-trained models, which are responsible for encoding images and texts into a meaningful sequence of tokens. Both models are required to be based on the transformer architecture (Vaswani et al., 2017), so we that can easily connect them with a few trainable components of similar designs. To preserve their rich domain-specific knowledge encoded in the network parameters, the majority of the weights are frozen during pre-training.

**Task Experts**  The models that produce multiple task-specific labels, depending on their training datasets, are treated as *black-box predictors.* These task experts can be designed either as a single multi-task expert or an ensemble of multiple task-specific experts, and their predicted labels are utilised as input for the Prismer model. Consequently, all network weights of the task experts are frozen, and they can have *any design.* In Prismer, we incorporate up to 6 task-specific experts, all within the vision domain. These experts encode three *low-level* vision signals (depth, surface normals, and edges) and three *high-level* vision signals (object labels, segmentation labels, and OCR labels). Our selection of these 6 vision experts is based on tasks commonly studied in the multi-task learning community (Zamir et al., 2018; Standley et al., 2020; Liu et al., 2022), which have demonstrated varying levels of benefits in learning generalised visual representations. Additionally, these expert models are relatively lightweight, incurring minimal additional training and inference costs with simple model parallelism.

We apply task-specific post-processing on these predicted labels, transforming them to a $\mathbb{R}^{H \times W \times C}$ tensor (here $H, W, C$ represent image height, width and channels respectively. *e.g.* $C = 1$ for depth and edge labels, and $C = 3$ for surface normals label). For all expert labels encoding high-level signals, we tile each pixel with its corresponding text embedding from a pre-trained CLIP text model (Radford et al., 2021), and then we apply PCA to down-sample the dimensionality to $C = 64$ for efficient training. The detailed descriptions of all task experts, including their pre-trained datasets and the architecture design, are listed in Appendix A.

## 3.3 Key Architectural Components

**Task-Specific Convolutional Stem**  All expert labels are first processed with randomly initialised convolution layers to map them to the same dimensionality. Specifically, we apply 5 convolutional layers and each is composed of a small $[3 \times 3]$ kernel, which is shown to perform better than a single convolutional layer but with a larger kernel in the original Vision Transformer design (Dosovitskiy et al., 2020), consistent with the finding in (Xiao et al., 2021). The convolutional stem is designed to be task-specific, which we have found to yield superior performance in comparison to a shared design in a multi-task learning setting (Liu et al., 2019b; Misra et al., 2016).

For high-level semantic labels such as those in object detection, semantic segmentation, and OCR detection, we down-sample the resolution by a factor of 4 to conserve running memory. Furthermore, for each object instance, we add a trainable and randomly sampled embedding to distinguish among different object instances. The size of this instance embedding is set to 128, which corresponds to the maximum possible number of object instances to be present in a single image. For RGB images, we simply process with the pre-trained convolutional stem defined by the original vision backbone. All task expert embeddings, including RGB, are then added with a pre-trained positional embedding before being further processed by transformer layers.

**Experts Resampler**  The computational complexity of self-attention is *quadratically proportional* to the number of input tokens. As such, the vision encoder can easily require tremendous memory when including a large number of task experts. To address this, we propose *Experts Resampler*, which takes a *variable* number

of expert labels as input and outputs a *fixed* number of tokens, illustrated in Fig. 3 Left. Such design produces a *constant* memory for the self-attention in the vision encoder, as well as the vision-text cross attention in the language decoder (shown in Fig. 2), independent of the inclusion of a different number of experts. Inspired by the design in the Perceiver Resampler (Jaegle et al., 2021) and the Flamingo model (Alayrac et al., 2022), the Experts Resampler learns a pre-defined number of latent input queries, to cross-attend a flattened multi-task features. The resampler then compresses the multi-task features into a much smaller number of tokens equal to the number of learned latent queries, as a form of *auxiliary knowledge distillation*. We design keys and queries to be a concatenation for both multi-task features and the learned latent queries, which is shown to be more effective, consistent with the design in the Flamingo model (Alayrac et al., 2022).

**Lightweight Adaptor** We insert one lightweight adaptor into each transformer layer of both vision and language backbones in order to improve Prismer's expressivity and conditioning on multi-task features, illustrated in Fig. 3 Right. The adaptor has an encoder-decoder design, which has proven to be successful for efficient transfer learning in the NLP domain (Houlsby et al., 2019; Pfeiffer et al., 2020). It first down-projects the input features into a smaller dimension, applies a non-linearity, and then up-projects the features back to the original input dimension. We choose the non-linearity function to be squared ReLU (So et al., 2021) – a simple and parameter-free function that delivers strong training stability. With the residual connection, we initialise all adaptors with near-zero weights to approximate the identity function. Combined with a standard cross

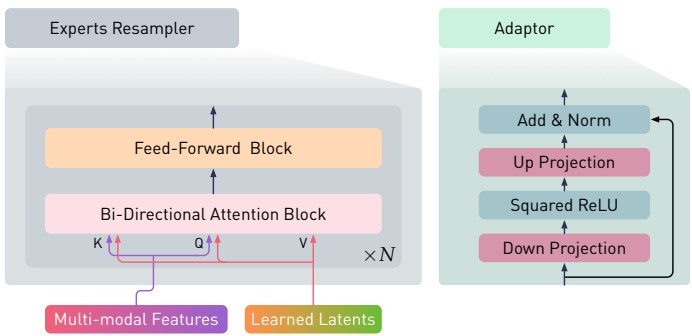

Figure 3: **Design details in Experts Resampler and Adaptor.** Left: The Experts Resampler takes multi-task features with variable length as input, and outputs a fixed number of tokens via cross attention. Right: The Adaptor has a residual connection to the input and two fully-connected layers, that down-projects the input features to a smaller bottleneck dimension and then up-projects back to the original dimension.

attention block in the language decoder, the model is able to smoothly transition from the domain-specific vision-only and language-only backbones to a vision-language model during pre-training with paired image-text data.

The model performance, memory usage and time complexity for other design choices are systematically evaluated and ablated in Appendix B.

### 3.4 Training Objective

For simplicity, we train Prismer with a *single* objective — to predict the next text autoregressively. Following the standard encoder-decoder architecture, the vision encoder predicts the multi-task features $z$, and the language decoder learns to maximise the conditional likelihood of the paired text caption $y$ under the forward autoregressive factorisation: $L = -\sum_{t=1}^{T} \log p(y_t | y_{<t}, z)$.

In practice, our *one-time* pre-processing step of collecting multi-task expert labels is computationally cheap and fast with data and model parallelism. The single generative objective then only requires one forward pass to compute gradients, which is significantly more efficient and streamlined than many other VLMs that may require a multi-stage and/or multi-step pre-training (Li et al., 2022; 2021; Wang et al., 2022b; Dou et al., 2022; Chen et al., 2020), with multiple objectives and data sources. However, because our model only focuses on multi-modal language generation, it is less suitable for multi-modal discriminative tasks such as image-text retrieval and visual entailment, which are the focus of other types of VLMs (Gan et al., 2020; Chen et al., 2020; Jia et al., 2021).

| | Resampler | | Vision Encoder | | | Language Decoder | | | Trainable Params. | Total Params. |
|---|---|---|---|---|---|---|---|---|---|---|
| | Layers | Width | Backbone | Layers | Width | Backbone | Layers | Width | | |
| Prismer$_{\text{BASE}}$ | 4 | 768 | ViT-B/16 | 12 | 768 | RoBERTa$_{\text{BASE}}$ | 12 | 768 | 160M | 980M |
| Prismer$_{\text{LARGE}}$ | 4 | 1024 | ViT-L/14 | 24 | 1024 | RoBERTa$_{\text{LARGE}}$ | 24 | 1024 | 360M | 1.6B |
| PrismerZ$_{\text{BASE}}$ | - | - | ViT-B/16 | 12 | 768 | RoBERTa$_{\text{BASE}}$ | 12 | 768 | 105M | 275M |
| PrismerZ$_{\text{LARGE}}$ | - | - | ViT-L/14 | 24 | 1024 | RoBERTa$_{\text{LARGE}}$ | 24 | 1024 | 270M | 870M |

Table 1: **Prismer and PrismerZ architecture details.** We report the backbone we choose for each architecture size, along with its corresponding number of layers and width. We also report the number of trainable parameters and total parameters for each architecture. We count the total parameters required for data inference, which include the additional 6 task experts with a combined parameter size of 654M parameters in our Prismer model.

# 4 Experiments

## 4.1 Prismer Model Variants

In addition to Prismer, we also introduce a model variant named PrismerZ, which solely relies on the power of strong backbone experts and is trained with *zero* task experts. PrismerZ has the same architectural design as the original Prismer but without the Experts Resampler. PrismerZ simplifies the data inference process as it only requires RGB images, making it more efficient and applicable to a wider range of applications. Prismer is less efficient in data inference due to the need for data processing on expert labels, but as we will show, it has better predictive performance.

Both Prismer and PrismerZ utilise ViT (Dosovitskiy et al., 2020) pre-trained by CLIP (Radford et al., 2021) as the frozen vision encoder, and RoBERTa (Liu et al., 2019c) as the frozen language decoder. We have also tried using two other popular open-sourced decoder-only autoregressive language models: OPT (Zhang et al., 2022) and BLOOM (Scao et al., 2022), but early experiments showed that they did not perform as well.

We experiment with two model sizes, BASE and LARGE. The BASE model is based on ViT-B/16 and RoBERTa$_{\text{BASE}}$, and the LARGE model is based on ViT-L/14 and RoBERTa$_{\text{LARGE}}$. In Prismer, we apply the same Experts Resampler with roughly 50M parameters in both model sizes. The architecture details are further summarised in Table 1.

## 4.2 Training and Evaluation Details

**Pre-training Datasets** We construct our pre-training data from the following datasets: two in-domain datasets: COCO (Lin et al., 2014) and Visual Genome (Krishna et al., 2017); and three web datasets: Conceptual Captions (Sharma et al., 2018), SBU captions (Ordonez et al., 2011), and a much noisier Conceptual 12M (Changpinyo et al., 2021). The web datasets are pre-filtered and re-captioned by a pre-trained image captioner (Li et al., 2022). The pre-training datasets include 11M unique images or 12.7M image/alt-text pairs.[2] All datasets are available publicly and have been widely used for pre-training many VLMs (Li et al., 2021; 2022; Chen et al., 2020).

**Optimisation and Implementation** All our models are trained with AdamW optimiser (Loshchilov & Hutter, 2019) with a weight decay of 0.05. Since only a small proportion of the model parameters are trainable, model sharding is only applied during fine-tuning on large-resolution images. Specifically, we employ ZeRO Stage 2 technique (Rajbhandari et al., 2020), which enables the sharding of optimiser states and parameter gradients across all GPU instances. Additionally, we also apply Automatic Mixed Precision (AMP) with fp16 precision to further reduce training time. For more details on our data processing techniques and hyper-parameter choices, please refer to Appendix B. An analysis of training and inference costs compared to other vision-language models can be found in Appendix C.

---

[2]This is slightly less than the theoretical number which should be 14M unique images. It is because some image URLs in the web datasets are not valid during the time we downloaded the datasets.

| | Pre-train (# Pairs) | COCO Caption | | | | NoCaps | | | | VQAv2 | |
|---|---|---|---|---|---|---|---|---|---|---|---|
| | | B @ 4 | M | C | S | In | Near | Out | Overall | test-dev | test-std |
| OSCAR_BASE (Li et al., 2020b) | 6.5M | 36.5 | 30.3 | 123.7 | 23.1 | 83.4 | 81.6 | 77.6 | 81.1 | 73.2 | 73.4 |
| VinVL_BASE (Zhang et al., 2021) | 8.9M | 38.2 | 30.3 | 129.3 | 23.6 | 103.7 | 95.6 | 83.8 | 94.3 | 76.0 | 76.1 |
| GIT_BASE (Wang et al., 2022a)† | 10M | **40.4** | 30.0 | 131.4 | 23.0 | 100.7 | 97.7 | 89.6 | 96.6 | 72.7 | - |
| BLIP_BASE (Li et al., 2022)† | 129M | 39.7 | - | 133.3 | - | **111.8** | **108.6** | 111.5 | **109.6** | **78.3** | **78.3** |
| LEMON_BASE (Hu et al., 2022) | 200M | 40.3 | 30.2 | 133.3 | 23.3 | 107.7 | 106.2 | 107.9 | 106.8 | - | - |
| PrismerZ_BASE† | 12.7M | 39.7 | 31.1 | 133.7 | 24.1 | 108.7 | 107.8 | 105.8 | 107.5 | 76.6 | - |
| Prismer_BASE† | 12.7M | 40.1 | **31.1** | **135.1** | **24.1** | 108.8 | 108.3 | **111.7** | 109.1 | 76.8 | 77.0 |
| OSCAR_LARGE (Li et al., 2020b) | 6.5M | 37.4 | 30.7 | 127.8 | 23.5 | 85.4 | 84.0 | 80.3 | 83.4 | 73.4 | 73.8 |
| VinVL_LARGE (Zhang et al., 2021) | 8.9M | 38.5 | 30.4 | 130.8 | 23.4 | - | - | - | - | 76.5 | 76.6 |
| GIT_LARGE (Wang et al., 2022a)† | 20M | **42.0** | 30.8 | **138.5** | 23.8 | 107.7 | 107.8 | 102.5 | 106.9 | 75.5 | - |
| BLIP_LARGE (Li et al., 2022)† | 129M | 40.4 | - | 136.7 | - | 114.9 | 112.1 | **115.3** | 113.2 | - | - |
| LEMON_LARGE (Hu et al., 2022) | 200M | 40.6 | 30.4 | 135.7 | 23.5 | **116.9** | **113.3** | 111.3 | **113.4** | - | - |
| PrismerZ_LARGE† | 12.7M | 40.0 | 31.2 | 135.7 | 24.2 | 112.3 | 111.2 | 112.8 | 111.8 | 77.5 | - |
| Prismer_LARGE† | 12.7M | 40.4 | **31.4** | 136.5 | **24.4** | 114.2 | 112.5 | 113.5 | 112.9 | **78.4** | **78.5** |
| LEMON_HUGE (Hu et al., 2022) | 200M | 41.5 | 30.8 | 139.1 | 24.1 | 118.0 | 116.3 | 120.2 | 117.3 | - | - |
| SimVLM_HUGE (Wang et al., 2021) | 1.8B | 40.6 | 33.7 | 143.3 | 25.4 | 113.7 | 110.9 | 115.2 | 112.2 | 80.0 | 80.3 |
| GIT (Wang et al., 2022a)† | 0.8B | 44.1 | 31.5 | 144.8 | 24.7 | 129.8 | 124.1 | 127.1 | 125.5 | 78.6 | 78.8 |
| GIT-2 (Wang et al., 2022a)† | 12.9B | 44.1 | 31.4 | 145.0 | 24.8 | 126.9 | 125.8 | 130.6 | 126.9 | 81.7 | 81.9 |
| CoCa (Yu et al., 2022) | 4.8B | 40.9 | 33.9 | 143.6 | 24.7 | - | - | - | 122.4 | 82.3 | 82.3 |
| PaLI (Chen et al., 2023)† | 1.6B | - | - | 149.1 | - | - | - | - | 127.0 | 84.3 | 84.3 |

Table 2: **Fine-tuned performance on COCO Caption (Karpathy split), NoCaps (validation set) and VQAv2.** Both Prismer and PrismerZ achieve superior performance in all three datasets compared to other VLMs with similar model sizes. Prismer can achieve competitive performance on par with VLMs that are trained with orders of magnitude more data. {B@4, M, C, S} refer to BLEU@4, METEOR, CIDEr, SPICE respectively. {In, Near, Out} refer to in-domain, near-domain and out-of-domain respectively. † evaluates the VQAv2 dataset in a generative setting. All other models evaluate the VQAv2 dataset in a closed-ended discriminative setting.

**Evaluation Setting** We evaluate the performance of our models through *generative language modelling*, which is a more challenging task than discriminative learning (particularly in VQA tasks), and aligns with that used in other vision-language generative models (Li et al., 2022; Alayrac et al., 2022; Wang et al., 2022a; Chen et al., 2023). For example, the model must accurately generate all text tokens for a question (which is on average 2.2 tokens per question in the VQAv2 dataset (Antol et al., 2015) as reported in (Wang et al., 2022a)), rather than just one correct prediction as required in discriminative models.

Specifically, we evaluate image captioning tasks in an open-ended generative setting, and we apply beam search with a beam size of 3 for text generation. A prefix prompt of "A picture of" is added to the input text for fined-tuned image captioning tasks, similar to previous studies in (Wang et al., 2021; Li et al., 2022; Radford et al., 2021), which have shown to improve the quality of image captions. We evaluate both VQA and image classification tasks in a close-ended generative setting, by ranking the per-token log-likelihood from a pre-defined answer list.

## 4.3 Results on Vision-Language Benchmarks

**Fine-tuned Performance on COCO Caption, NoCaps and VQAv2** We fine-tune our models on COCO Caption dataset (Chen et al., 2015) on a widely adopted Karpathy split (Karpathy & Fei-Fei, 2015), with the standard cross-entropy loss, and without metric-specific optimisation (Vedantam et al., 2015). We evaluate the fine-tuned models on the COCO Caption Karpathy test split and NoCaps (Agrawal et al., 2019) validation set. We also evaluate our models on the VQAv2 dataset (Antol et al., 2015), with additional training samples from Visual Genome (Krishna et al., 2017) following (Li et al., 2022). We compare our models with prior state-of-the-art VLMs that are mostly pre-trained on image-text data for a fair comparison. We sort all VLMs by their model sizes and report the results in Table 2.

The results show that both Prismer and PrismerZ achieve superior performance considering their model sizes, which suggests that the strong backbone experts are primarily responsible for good generalisation. However, the task experts provide an additional boost in performance, particularly in image captioning tasks

|  | NoCaps | |
| --- | --- | --- |
|  | C | S |
| FewVLM (Jin et al., 2022) | 47.7 | 9.1 |
| MetaLM (Hao et al., 2022) | 58.7 | 8.6 |
| VLKD (Dai et al., 2022) | 63.6 | 12.8 |
| SimVLM_HUGE (Wang et al., 2021) | 101.4 | - |
| BLIP-2 (Vicuna-7B) (Li et al., 2023) | 107.5 | - |
| BLIP-2 (Vicuna-13B) (Li et al., 2023) | 103.9 | - |
| Prismer_BASE | 87.5 | 13.0 |
| Prismer_LARGE | 107.9 | 14.8 |

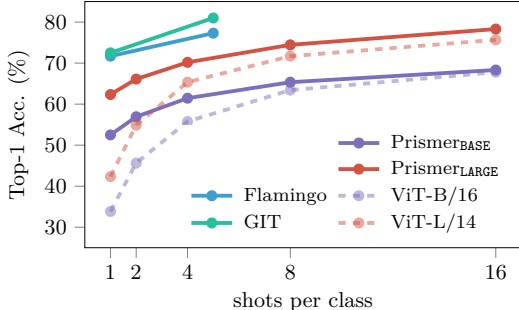

Figure 4: **Results on zero-shot image captioning and few-shot ImageNet classification.** Left: Prismer achieves state-of-the-art zero-shot image-captioning results on NoCaps (validation set), with similar performance to SimVLM and BLIP-2, despite being trained on 140 times and 10 times less data respectively. Right: Prismer significantly improves few-shot performance compared to its corresponding vision backbone. However, Prismer still underperforms GIT and Flamingo which are trained on significantly more data.

(such as a 6 CIDEr score increase in the NoCaps out-of-domain set in the BASE model) and in the LARGE model variant (such as a 1 VQAv2 accuracy increase in the LARGE model). Both Prismer_BASE and Prismer_LARGE achieve comparable image captioning performance to BLIP (Li et al., 2022) and LEMON (Hu et al., 2022), despite being trained on 10 and 20 times less data, respectively. Additionally, the Prismer_LARGE model has achieved VQAv2 accuracy comparable to GIT (Wang et al., 2022a), despite being trained on 60 times less data. Whilst we acknowledge a noticeable performance gap between Prismer and the current state-of-the-art VLMs (such as CoCa (Yu et al., 2022), GIT-2 (Wang et al., 2022a) and PaLI (Chen et al., 2023)), these models require substantially higher training costs and access to large-scale private training data.

**Zero-shot Performance on Image Captioning** Our generative pre-training approach allows for zero-shot generalisation, where the models can be directly applied to image captioning tasks without additional fine-tuning. In Fig. 4 Left, we show that Prismer achieves state-of-the-art performance on the NoCaps dataset similar to SimVLM (Wang et al., 2021) and BLIP-2 (Li et al., 2023), whilst using significantly smaller network parameter size and trained with 140 times and 10 times less data respectively. Additionally, we notice that the zero-shot performance of Prismer models even surpasses the fine-tuned performance of certain VLMs such as OSCAR (Li et al., 2020b) and VinVL (Zhang et al., 2021), as shown in Table 2.

We present a list of example captions generated by Prismer in Table 3. The results show that both Prismer_BASE and Prismer_LARGE are capable of generating captions that are semantically coherent and aligned with the visual content of the images. Notably, Prismer_LARGE generates captions of higher quality compared to Prismer_BASE, exhibiting a deep understanding of fine-grained object semantics such as brand recognition (*e.g.* Mercedes, CK One), and cultural concepts (*e.g.* vintage drawing, tango), indistinguishable to human-written captions.

**Few-shot Performance on ImageNet Classification** Finally, we fine-tune and evaluate Prismer on ImageNet (Deng et al., 2009) in a few-shot setting. Following the approach outlined in (Radford et al., 2021), we convert the classification task into a language modelling problem by mapping each category to a template caption: "A photo of a [CLASS NAME]", and we then score captions using the log-likelihood estimated by our model. Unlike Flamingo (Alayrac et al., 2022) which performs few-shot classification via in-context examples without gradient updates, we perform few-shot classification via lightweight fine-tuning following (Wang et al., 2022a). This is more similar to the standard linear probe setting, by considering the entire language decoder as an image classifier. Accordingly, we also compare with the few-shot linear probe performance of Prismer's original vision backbones ViT-B/16 and ViT-L/14 (Dosovitskiy et al., 2020), as reported in (Schuhmann et al., 2022; Radford et al., 2021).

From the results shown in Fig. 4 Right, we observe that Prismer underperforms GIT (Wang et al., 2022a) and Flamingo (Alayrac et al., 2022), which both have stronger and larger vision backbones and are pre-trained on

| | Ground-Truth | PrismerBASE | PrismerLARGE |
|---|---|---|---|
|  | 1. A clear bottle of CK cologne is full of liquid. 

 2. The bottle of perfume is made by Calvin Klein. | A bottle of alcohol sitting next to a computer keyboard. | A bottle of ck one next to a computer keyboard. |
|  | 1. A statue has a large purple headdress on it. 

 2. A woman decorated in fashioned clothing and relics. | The woman is wearing a black dress. | A mannequin dressed in a black dress with feathers on her head. |
|  | 1. A new white car with the door open is in a showroom full of people. 

 2. A shiny white mercedes car is on display. | A white car on display at a car show. | A white mercedes car on display at an auto show. |
|  | 1. Large piece of meat with slices of pineapple with cherries being held on with toothpicks on blue and white plate. 

 2. A cake has several slices of pineapple and cheries in them. | Pineapples on a plate. | Pineapple upside down cake on a blue and white plate. |
|  | 1. A man and woman is dancing as a crowd watches them in the distance. 

 2. A woman in a red dress dancing with a bald man wearing black. | A couple of people that are standing in the dirt. | A couple dancing tango in front of a crowd. |
|  | 1. Two illustrations of lobster colors are shown as Fig. 21 and Fig. 22. 

 2. A drawing of a lobster and a lobster. | Colored drawing of two lobsters on pink paper. | A vintage illustration of lobsters from the 19th century. |
|  | 1. Man in skydiving gear giving two thumbs up with skydivers in the sky behind him. 

 2. Person giving double thumbs up sign while others are parachuting in the background. | Man wearing a blue and purple jacket. | A man wearing a helmet and goggles with parachutes in the background. |

Table 3: **Visualisation of zero-shot image captioning on NoCaps.** PrismerLARGE produces more detailed and semantically coherent captions than PrismerBASE, showing an understanding of fine-grained object recognition and abstractions. Results are *not* cherry-picked.

significantly more data. However, Prismer still outperforms its original vision backbones ViT-B and ViT-L by a large margin, especially in a very few-shot setting, despite having the exact same representation space. This suggests that Prismer's generalisation abilities are enhanced by the multi-modal training data and expert labels, and its performance can likely be improved further by using an even stronger vision backbone.

## 5 Additional Analysis

We now include a comprehensive evaluation of Prismer, characterised by a meticulous and fine-grained analysis of its learning strategy. We delve into various aspects of Prismer's performance, examining its behaviour with different types of multi-task experts (as discussed in Sec.5.1). Additionally, we explore the individual utility of each expert in addressing domain-specific reasoning tasks, allowing us to gain insights into the specific strengths and contributions of each expert (as discussed in Sec.5.2).

### 5.1 Intriguing Learning Strategy of Prismer

To speed up training, all experiments are conducted with the `BASE` model on a combined dataset of the Conceptual Captions and SBU, consisting of a total of 3M data. All experiments are evaluated on the VQAv2 `test-dev` split in a smaller $[224 \times 224]$ resolution.

**More Experts, Better Performance** We observe that the performance of Prismer improves with more task experts, as shown in Fig. 5a. This is intuitive because more experts provide a greater diversity of domain knowledge to the model. However, we also note that the performance of the model eventually plateaus, which suggests that additional task experts beyond a certain number do not provide any extra gains.

**Better Experts, Better Performance** To evaluate the impact of expert quality on Prismer's performance, we construct a *corrupted* depth expert by replacing a certain number of predicted depth labels with random noise sampled from a Uniform Distribution. As shown in Fig. 5b, Prismer's performance improves as the quality of the depth expert improves. This is intuitive as better experts provide more accurate domain knowledge, allowing the model to perceive more accurately.

**Robustness to Noisy Experts** Our results also demonstrate that Prismer maintains performance even when including experts that predict noise, as shown in Fig. 5c. Interestingly, adding noise can even result in a non-trivial improvement compared to training on RGB images alone, which can be considered as a form of implicit regularisation. This property allows the model to safely include many experts *without degrading the performance*, even when the expert is *not necessarily informative*. Therefore, Prismer presents a more effective learning strategy than the standard multi-task or auxiliary learning methods, which either require exploring task relationships (Liu et al., 2022; Fifty et al., 2021; Zamir et al., 2018) or designing more advanced optimisation procedures (Liu et al., 2019a; Navon et al., 2021).

### 5.2 Utility of Task Experts

In this experiment, we conduct a comprehensive evaluation to assess the utility of each task expert within Prismer concerning different types of reasoning tasks. To accomplish this, we employ Prismer$_{\text{LARGE}}$, which was trained on the VQAv2 dataset, and evaluate its zero-shot performance in combination with each individual task expert on two specific domain-specific reasoning tasks: i) Visual Spatial Reasoning (VSR) (Liu et al., 2023): This task evaluates a VLM's spatial reasoning ability. It involves classifying image-caption pairs as either true or false, indicating whether the caption correctly describes the spatial relation in an image. ii) Text-VQA (Singh et al., 2019): This task assesses a VLM's ability to understand and reason about text within an image. It involves comprehending and answering questions related to text in an image.

The results presented in Table 4 demonstrate that Prismer consistently outperforms several competitive baselines, such as VisualBERT (Li et al., 2020a), LXMERT (Tan & Bansal, 2019), and ViLT (Kim et al., 2021) in the VSR dataset, all without requiring dataset-specific fine-tuning as required by these methods. Prismer

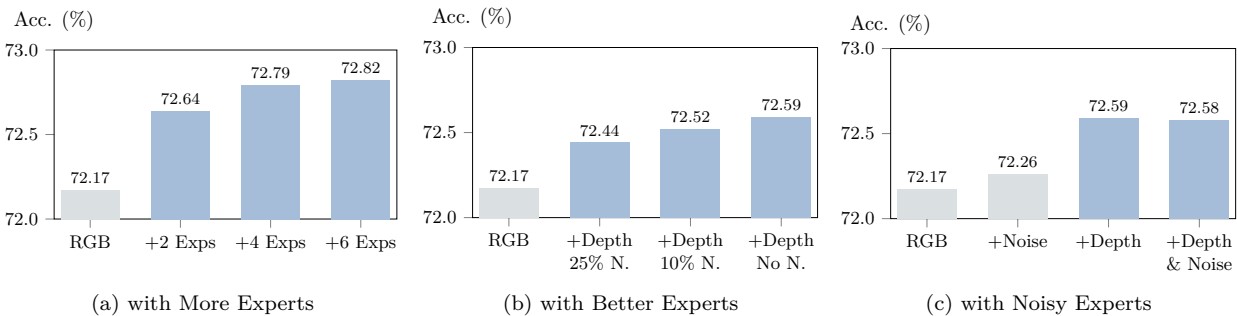

Figure 5: **Prismer's VQAv2 accuracy with different types and the number of experts.** Prismer has shown that its performance improves with an increase in the number and quality of task experts. Additionally, Prismer also demonstrates its strong robustness to noisy experts, making it a practical and effective multimodal learning strategy.

| Baselines (fine-tuned) | | | Prismer (zero-shot) | | | | | | | |
|---|---|---|---|---|---|---|---|---|---|---|
| VisualBERT | LXMERT | ViLT | +Depth | +Normal | +Edge | +Seg. | +OCR Det. | +Obj. Det. | No Experts | +6 Experts |
| 51.0 | 61.2 | 63.0 | 68.4 | 68.3 | 67.8 | 68.4 | 67.2 | 68.3 | 65.6 | 68.7 |

(a) VSR

| Baselines (zero-shot) | | | Prismer (zero-shot) | | | | | | | |
|---|---|---|---|---|---|---|---|---|---|---|
| OFA | BLIP-2 | Flamingo | +Depth | +Normal | +Edge | +Seg. | +OCR Det. | +Obj. Det. | No Experts | +6 Experts |
| 18.3 | 15.7 | 35.0 | 27.4 | 28.0 | 28.2 | 27.8 | 28.4 | 28.4 | 22.6 | 28.8 |

(b) Text-VQA

Table 4: **Zero-shot accuracies in VSR (zero-shot split) and Text-VQA (validation split) datasets, considering various types of experts.** These results shed light on the valuable contributions of individual experts for domain-specific reasoning tasks, offering insights into the versatility and adaptability of Prismer across different domains and problem-solving scenarios. The colour green represents the most helpful experts, while the colour red represents the least helpful experts.

also surpasses BLIP-2 [OPT 2.7B] (Li et al., 2023) and OFA$_{\text{HUGE}}$ (Wang et al., 2022b), despite employing a smaller backbone network and significantly less pre-training data respectively.

Furthermore, Prismer's utility analysis offers valuable insights into the contributions of individual experts in addressing specific reasoning tasks. For example, the "object detection" expert is identified as crucial in both the VSR and Text-VQA tasks, highlighting the significance of object recognition capability in general visual reasoning problems. Additionally, the "depth" and "OCR detection" experts are recognised as key contributors to Prismer's performance in spatial reasoning and text reasoning, respectively, aligning with human intuition — depth information enhances 3D spatial understanding, whilst OCR detection directly improves text reading capability.

Finally, the substantial performance improvement observed (compared to general reasoning tasks presented in Table 2) when comparing Prismer to PrismerZ (with no experts) underscores the pivotal role played by the experts in domain-specific reasoning tasks. This highlights the tangible benefits of incorporating experts within the Prismer architecture, particularly when tackling tasks that require specialised knowledge and reasoning capabilities.

# 6 Conclusions, Limitations and Discussion

In this paper, we have introduced Prismer, a vision-language model designed for reasoning tasks. Prismer is parameter-efficient and utilises a small number of trainable components to connect an ensemble of diverse, pre-trained experts. By leveraging these experts, Prismer achieves competitive performance in image captioning, VQA, and image classification benchmarks, comparable to models trained on up to two orders of magnitude

more data. For full transparency, we now discuss some limitations of Prismer during our implementation and explore potential future directions for this work.

**Multi-modal In-context Learning** Zero-shot in-context generalisation is an emergent property that only exists in very large language models (Brown et al., 2020; Wei et al., 2022). In this work, we build Prismer on top of a small-scale language model with the main focus on parameter-efficient learning. Therefore, it does not have the ability to perform few-shot in-context prompting by design.

**Zero-shot Adaptation on New Experts** We experiment with inference on a pre-trained Prismer with a different segmentation expert pre-trained on a different dataset. Although we apply the same language model to encode semantic labels, Prismer shows limited adaptability to a different expert with a different set of semantic information, which leads to a notable performance drop.

**Free-form Inference on Partial Experts** Similarly, we discover that Prismer entangles its multi-task features from all experts we include during pre-training. Therefore, only having a partial number of experts during inference will lead to a notable performance drop. We attempt to use a different training objective such as masked auto-encoding (Bachmann et al., 2022), to design Prismer to reason on an arbitrary number of experts, but it eventually leads to a degraded fine-tuned performance.

**Representation of Expert Knowledge** In our current design of Prismer, we convert all expert labels into an image-like 3-dimensional tensor via task-specific post-processing for simplicity. There are other efficient methods to represent expert knowledge, such as converting object detection into a sequence of text tokens (Chen et al., 2021; 2022). This may lead to stronger reasoning performance and a more stable training landscape in future works.

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

# A    Detailed Description of Experts

In Table 5, we present a detailed list of task experts we have included in the Prismer model. We report each expert's pre-trained domain, as well as its pre-trained dataset, parameter size, and post-processing techniques.

| Task | Dataset | Model | Params. | Post-Processing |
|---|---|---|---|---|
| Semantic Segmentation | COCO-Stuff (Caesar et al., 2018) | Mask2Former (Cheng et al., 2022) | 215M | In-paint each pixel with its corresponding label parametrised by CLIP language embedding. |
| Object Detection | COCO (Lin et al., 2014) + Objects365 (Shao et al., 2019) + OpenImages (Kuznetsova et al., 2020) + Mapillary (Neuhold et al., 2017) | UniDet (Zhou et al., 2022) | 120M | In-paint each pixel with its corresponding label parametrised by CLIP language embedding. The labels for the overlapping pixels are further determined by the depth expert. |
| Text Detection | ICDAR 2015 (Karatzas et al., 2015) | CharNet (Liu et al., 2018) | 89M | In-paint each pixel with its corresponding text parametrised by CLIP language embedding. |
| Depth Estimation | MIX-6 (Ranftl et al., 2021) | DPT (Ranftl et al., 2021) | 123M | Re-normalised to $[-1, 1]$. |
| Surface Normal | ScanNet (Dai et al., 2017) | NLL-AngMF (Bae et al., 2021) | 72M | Re-normalised to $[-1, 1]$. |
| Edge Detection | BIPED (Poma et al., 2020) | DexiNed (Poma et al., 2020) | 35M | Re-normalised to $[-1, 1]$. |

Table 5: **The detailed description of task experts**. We provide a detailed description of each task expert including its pre-trained dataset, parameter size, model name and type and its post-processing strategy.

# B    Detailed Training Strategy and Hyper-parameters

All Prismer model variants are pre-trained with a resolution of $[224 \times 224]$ and evaluated on three types of vision-language reasoning tasks: image captioning, visual question answering, and image classification. To improve performance, the models are fine-tuned with larger resolutions: $[384 \times 384]$ for image classification, and $[448 \times 448]$ for image captioning and visual question answering tasks. Automated data augmentation (Cubuk et al., 2020) is applied during both pre-training and fine-tuning. A list of the hyper-parameters used in the experiments can be found in Table 6.

For language generation, beam search is applied with a beam size of 3 for all image captioning datasets. Per-token log-likelihood is used to rank answers for the VQAv2 and ImageNet datasets. A prefix prompt of `A picture of` is added to the input text for image captioning tasks, similar to previous studies such as in (Wang et al., 2021; Li et al., 2022; Radford et al., 2021), as shown to improve the quality of image captions.

|  | Pre-training | COCO / NoCaps | VQAv2 | ImageNet |
|---|---|---|---|---|
| Optimiser |  | AdamW |  |  |
| LR Schedule |  | Cosine annealling to zero |  |  |
| Weight Decay |  | 0.05 |  |  |
| Warmup Steps | 2000 | 0 | 0 | 0 |
| Initial LR | $3/1 \cdot 10^{-4}$ (B / L) | $5 \cdot 10^{-5}$ | $5 \cdot 10^{-5}$ | $5 \cdot 10^{-5}$ |
| Resolution | 224 | 480 | 480 | 384 |
| Epochs | 20 | 3 | 10 | 20 |
| Batch Size | 1024 | 256 | 512 | 64 |

Table 6: **The detailed list of hyper-parameters and training strategy.** To ensure reproducibility, we have included a list of all hyper-parameters used in our experiments. These same hyper-parameters are applied to both the `BASE` and `LARGE` model variants.

# C    Comparison of Training Cost

Prismer is highly efficient in terms of training and inference costs. Here, the training costs are defined by the costs exclusively involved in constructing the Prismer models, excluding all original pre-training data implied in the original expert models. This definition aligns with the conventions in the VLM community. The largest model variant, Prismer$_{LARGE}$, only requires 8 days of training on 32 NVIDIA V100 GPUs. This is significantly more efficient than previous state-of-the-art VLMs such as SimVLM (Wang et al., 2021) which requires 5 days of training on 2048 TPUv3, GIT-2 (Wang et al., 2022a) which requires 1.5 months of training on 1500 NVIDIA A100s, and Flamingo (Alayrac et al., 2022) which requires 2 weeks of training on 1536 TPUv4. A detailed breakdown of the pre-training and inference costs can be found in Table 7.

|  | Model Params. | Pre-training Data (# Image-Text Pairs) | Pre-training Cost (# PFlops Days) | Inference Cost (# TFlops) |
|---|---|---|---|---|
| BLIP$_{LARGE}$ | 583M | 129M | 22.2[‡] | 0.17 |
| SimVLM$_{HUGE}$ | 1.4B | 1.8B | 66.9[‡] | 0.40 |
| GIT | 681M | 0.8B | 45.8[‡] | 0.37 |
| PaLI | 17B | 2.3B | 450 | 5.7 |
| Flamingo | 80B | 2.3B | 1.4K[†] | 23 |
| GIT-2 | 5.1B | 12.9B | 5.5K[†] | 2.6 |
| Prismer$_{BASE}$ | 980M | 12.7M | 0.66 | 0.20 |
| Prismer$_{LARGE}$ | 1.6B | 12.7M | 1.9 | 0.38 |

Table 7: **Training and inference cost of vision-language models.** We compare the training and inference cost of Prismer with several other vision-language models using the approximation method from (Brown et al., 2020). The symbol † represents the training cost estimated by (Chen et al., 2023), and ‡ represents the training cost estimated by us. All inference costs are estimated by us with an input of 256 image tokens and 30 text tokens.

## D   Ablation Study on Architecture Design and Training Details

To perform the ablation studies, we use the Prismer$_{\text{BASE}}$ model and train it on the Conceptual Captions and SBU with a total of 3M training data. The results of the ablation studies are presented in Table 8.

| Ablated Component | Our Setting | Changed Setting | Params. (Rel.) | Step Time (Rel.) | VQAv2 (Acc.) |
|---|---|---|---|---|---|
| **Prismer$_{\text{BASE}}$ (our setting with reduced training)** | | | **1.00** | **1.00** | **72.79** |
| (i) Adapter Design | Residual MLP | Residual MLP ×2 | 1.04 | 1.02 | 72.36 |
| | | Gated Residual MLP | 1.03 | 1.03 | 70.54 |
| (ii) Adapter Bottleneck Dim. | 1 | 1/2 | 0.95 | 0.96 | 72.52 |
| | | 1/4 | 0.93 | 0.93 | 71.66 |
| (iii) Resampler Design | Experts Perceiver | Random Sampling | 0.91 | 0.96 | 72.24 |
| | | Full Perceiver | 1.00 | 0.90 | 65.05 |
| | | Dual Perceiver | 1.08 | 1.02 | 71.56 |
| (iv) Resampler Layers | 4 | 1 | 0.94 | 0.93 | 70.61 |
| | | 2 | 0.96 | 0.96 | 72.39 |
| | | 6 | 1.04 | 1.01 | 72.78 |
| (v) Resampler Latents | 64 | 32 | 1.00 | 0.95 | 72.44 |
| | | 128 | 1.00 | 1.01 | 70.28 |
| | | 256 | 1.00 | 1.06 | 68.07 |
| (vi) Pre-training | Freeze Vision and Lang. | Freeze Vision Only | 1.00 | 1.07 | 70.49 |
| | | Freeze Lang. Only | 1.00 | 1.05 | 67.77 |
| | | All Parameters | 1.00 | 1.15 | 68.13 |
| (vii) Fine-tuning | Freeze Vision | Freeze Vision and Lang. | 1.00 | 1.00 | 71.36 |
| | | Freeze Lang. Only | 1.00 | 1.00 | 70.37 |
| | | All Parameters | 1.00 | 1.00 | 68.69 |

Table 8: **Ablation studies for architecture components and training strategies.** We perform ablation studies to evaluate the impact of different architectural components and training strategies on the VQAv2 `test-dev` performance. We compare the performance of our default setting to other design and training options. The number of parameters and pre-training step time of the changed setting relative to the default setting are reported. To ensure a fair comparison, all experiments are evaluated using a reduced amount of training data and 3 task experts: depth, normal and segmentation.

**Adaptor Design: Single and Simple**   In our ablation study of adaptor designs, as shown in row (i) and (ii) of Table 8, we find that the simplest adaptor design, which consisted of a standard residual connection and an encoder-decoder structure, performs the best. We have experimented with more complex designs, such as adding an additional adaptor at the end of each transformer layer or incorporating a learnable gating mechanism similar to that in (Liu et al., 2021), but both have achieved worse performance. We also observe that having a larger bottleneck hidden size for the single adaptor improves performance.

**Resampler Design: Auxiliary Knowledge Distillation**   In our ablation study of Experts Resampler designs and different sampling strategies for encoding multi-task signals, as shown in row (iii) - (v) of Table 8, we find that keeping the number of resampler layers and latents lightweight is essential for a stable training process. We also experiment with replacing the resampler with a non-learnable random sampling approach, which results in slightly worse performance compared to using the resampler. We attempt to make the resampler more efficient by receiving full signals, including the RGB, before self-attention, but this has resulted in significantly degraded performance. Additionally, we have tried adding an additional resampler at the end of the vision encoder, but this design also results in worse performance.

**Frozen Backbone to Preserve Web-Scale Knowledge**   In our experiments on pre-training and fine-tuning whilst freezing different parts of the model, as shown in row (vi) and (vii) of Table 8, we find that freezing pre-trained parameters is essential for achieving strong performance and avoiding over-fitting and

catastrophic forgetting of the learned web-scale knowledge.[3] Freezing these parameters has also saved a significant amount of GPU memory. Even when fine-tuning on different downstream tasks, we find that freezing the vision encoder is beneficial (whilst keeping the resampler and adaptors trainable). This observation is consistent with the findings in (Zhai et al., 2022), which shows that only fine-tuning the language model with a frozen vision model for vision-language contrastive learning can achieve much stronger zero-shot performance.

---

[3]We assume the size of our pre-training data is significantly smaller than the original pre-training data used to train these backbone models.

