# OpenReview forum: "Prismer: A Vision-Language Model with Multi-Task Experts"
_TMLR — Accepted by TMLR_

### Review · Reviewer_cr8h · 2023-11-14

**Summary Of Contributions:**

The paper introduces the Prismer model, which relies on a set of pre-trained models referred to as "experts." These experts leverage diverse visual information, encompassing low-level signals like depth and high-level signals such as objects and semantic labels. The auxiliary knowledge is extracted directly from the network outputs of these individually pre-trained and frozen expert models. Through the connection of these experts using lightweight trainable components, Prismer achieves reduced computational costs and data requirements.

**Audience:**

No

**Claims And Evidence:**

No

**Requested Changes:**

-	The using of task experts just brings a little improvement in some specific tasks/datasets. The motivation of the proposed approach could be clearer. What happens if the selected separate tasks are not available or poorly trained?
-	Given the use of cross-attention in the expert resampler, could the authors shed light on how much attention the model allocates to specific experts? Additionally, an explanation of why a particular expert proves beneficial while others do not would enhance understanding.
-	The paper selects six task experts, and it would be valuable to understand the intuition behind these choices.
-	To facilitate easier comprehension for readers, the authors should include the comparison of pre-training cost in the results table.

**Strengths And Weaknesses:**

Strengths:

-	The paper is well-presented and easy to follow.

-	The proposed method offers an alternative approach for a single model capable of multiple tasks by ensembling available pre-trained models on separate tasks. It has the advantage of being less computationally expensive and smaller in size in comparison with large-scale pre-trained models.

Weakness:

-	Although the authors illustrate that task experts contribute to some improvement, the enhancements shown in Table 2 seem relatively marginal. The primary contribution of the paper lies in leveraging pre-trained models on separate tasks; however, it appears to be limited in terms of novelty.

-	There is a lack of in-depth analysis regarding the specific impact of each individual task on the rest.

---

> ### Author Response · Authors · 2023-11-21
> **Response 1/2**
>
> Thanks for highlighting the importance and clarity of our research. Here, we resolve each of your concerns below.
>
> >The using of task experts just brings a little improvement in some specific tasks/datasets.
>
> While we acknowledge the concerns regarding the significance of performance improvement in some selected tasks (most likely referring to the fine-tuned performance in Table 2), we respectfully contend that performance improvement is not trivial. It is essential to underscore that VQAv2 (along with other captioning datasets) is characterised by rigorous competition and is instrumental in evaluating the model's advancements.
>
> Simply scaling training data and model size would result in similar or even less performance improvement. For example, increasing the language model from BERT-BASE to BERT-LARGE would only increase OSCAR from 73.2 to 73.4 (+0.2\%) and VinVL from 76.0 to 76.5 (+0.5\%); increasing the vision backbone from ViT-BASE to ViT-LARGE and ViT-HUGE in SimVLM would result (77.5 -> 79.3 -> 80.0, +1.8\% and +0.7\%). In particular, we would like to highlight BLIP, a generative VLM that shares the exact same performance as PrismerZ at 77.5 accuracy. **By augmenting the training data by a factor of 10 (14M -> 129M) escalates BLIP's performance to 78.3 accuracy. **Contrarily, **Prismer manages to elevate PrismerZ with a similar amount of performance (77.5 -> 78.4) without additional data**, emphasising the salient contribution of multi-task experts, and further confirming that the performance improvements are non-trivial.
>
> Finally, as the reviewer also has noticed, **the strongest benefits of Prismer exhibit in few/zero-shot learning settings**, as depicted in Figure 4 and Table 4. Notably, Prismer achieves state-of-the-art performance in the zero-shot NoCaps dataset and domain-specific reasoning datasets, surpassing even the recently introduced BLIP-2, despite being built on top of a substantially smaller language model with 10x less parameter size. In such a setting, including task-experts has led to a significant improvement (22.6 -> 28.8, 25\% relative improvement in zero-shot Text-VQA).
>
> >The motivation of the proposed approach could be clearer. What happens if the selected separate tasks are not available or poorly trained?
>
> Thank you for raising this intriguing question. **We have addressed this analysis in Section 5.1**, wherein we demonstrate that Prismer exhibits **robustness to noisy experts** that simply generate random noise or provide useless labels.  This design feature underscores the importance of Prismer **considering multi-task experts as auxiliary knowledge rather than relying solely on their labels**. This sets Prismer apart from other VLMs such as Visual Programming and ViperGPT, which depend on multi-task experts for sequential reasoning. In those models, incorrect predictions from any individual experts can lead to model failure in accurately answering questions.
>
>
> >Given the use of cross-attention in the expert resampler, could the authors shed light on how much attention the model allocates to specific experts? Additionally, an explanation of why a particular expert proves beneficial while others do not would enhance understanding.
>
> We greatly acknowledge the importance of understanding the Prismer's reasoning strategy within the context of each task expert. However, we would like to argue that the attainment of such interpretability, unfortunately, necessitates a trade-off stemming from the compact architecture integrating the Expert Resampler.
>
> In our original Prismer design, the absence of the Resampler allowed for an analysis of each expert's contribution through its corresponding relative attention scores for the predicted answers, as suggested by the reviewer. However, this design faced scalability limitations concerning the number of experts, resulting in a quadratic increase in the memory compute within the vision encoder. To address this challenge, we devised the present solution with the Expert Resampler. This component effectively limits the expansion of the multi-modal experts with a fixed token length, but comes at the cost of sacrificing interpretability -- i.e. **the compressed learned latents do not have a one-to-one correspondence to each task expert**. To address this trade-off, we conducted a thorough analysis in Section 5.2 by performing evaluations with the inclusion of each individual expert for domain-specific reasoning. Our findings indicate that **the most helpful experts align with human intuition** (OCR detection helps most on Text-VQA; and depth estimation helps most on spatial reasoning). We acknowledge and plan to emphasise this limitation in the final version of our paper.

---

> ### Author Response · Authors · 2023-11-21
> **Response 2/2**
>
> >The paper selects six task experts, and it would be valuable to understand the intuition behind these choices.
>
> Given that the design of the Expert Resampler provides constant multi-task latents independent of the number of experts, it allows for the straightforward addition of as many experts as desired. In our implementation, we have chosen **six experts, each representing distinct tasks commonly evaluated and developed in the vision and multi-task learning community** (UberNet (Kokkinos et al., 2016), MTAN (Liu et al., 2019), and Taskonomy (Zamir et al., 2018)).
>
> >To facilitate easier comprehension for readers, the authors should include the comparison of pre-training cost in the results table.
>
> We acknowledge the importance of including pre-training costs, as it facilitates a more accessible comparison with other baseline models. In our paper, we have **presented a table in Appendix C** that explicitly demonstrates Prismer's capability to achieve **10 to 50 times greater training efficiency compared to other VLMs** while maintaining similar performance levels.
>
> -----------
> Request Changes:
> 1. We have updated the explanation for the selection of task experts.
> 2. The role and robustness of each expert have already been included in the experiments (Section 5.1 and 5.2).
> 3. Inference costs have been added; training costs are already included in the paper (Appendix C).

---

> ### Author Response · Authors · 2023-12-18
> **Paper Update**
>
> Dear Reviewer,
>
> We have thoroughly revised the paper, addressing the comments and concerns you raised. A detailed response to your concerns and comments is provided above. We would appreciate your feedback on whether these updates sufficiently address your comments.
>
> Thank you!
> Authors

---

### Review · Reviewer_8Dhj · 2023-11-20

**Summary Of Contributions:**

The Prismer model proposed by the paper fuses the outputs from a pool of pretrained experts for vision-language reasoning. The author adopted a transformer encoder-decoder architecture for expert signal fusion, and initialized the encoder with ViT and decoder with RoBERTa. The author also inserted adapters in the network and only finetunes the adapters in Prismer. Experiments show that 1) Prismer achieves state-of-the-art zero-shot image captioning results in NoCaps (as suggested by Figure 4), 2) Prismer is robust to noisy experts, i.e., adding experts that are not necessarily informative won't degrade the performance, 3) the performance of Prismer gets better when there are more experts.

**Audience:**

Yes

**Claims And Evidence:**

Yes

**Requested Changes:**

The author needs to 1) compare with Unified-IO, 2) conduct more ablation study on the design of adapters. See Strengths And Weaknesses for the whole list of weaknesses.

**Strengths And Weaknesses:**

Strengths:

1. The idea of tackling vision-language reasoning by ensembling models trained on different tasks is interesting. Compared with solutions that train from scratch or finetune the vision-language model, Primser is considerably cheaper but still have reasonable performance.
2. Experiments show that Prismer is robust to noise in the experts and gets better when there are more experts.

Weaknesses:

1. The author has not mentioned some important related works, like Unified-IO (https://arxiv.org/pdf/2206.08916.pdf). Unified-IO also casts a bunch of vision-language tasks as sequence-to-sequence learning problems and trained a unified model that can solve all tasks. Different from Prismer, Unified-IO does not depend on the output from pretrained experts. Thus, the author should also compare with Unified-IO.
2. The author indicated in the limitation section that Prismer is not good at adapting existing experts to new experts. This shows that performance of Prismer is tied to the choice of the experts, which impairs Prismer's flexibility.
3. As shown in the leftmost part of Figure 5, The accuracy boost obtained by adding more experts is diminishing. Prismer with 6 experts is 0.03 better than Prismer with 4 experts. Moreover, the right part of Figure 4 shows that the performance of Prismer falls significantly behind Flamingo and GIT.
4. There are not enough ablation study that justifies the design of the adapter. For example, will the model perform worse or better with full-finetuning? In fact, the author mentioned that Prismer is trained on significantly less amount of data than Flamingo / GIT. It is known that full fine-tuning works better than PEFT if we have a large number of data samples. The author can thus compare the performance difference between Prismer + full finetuning and Prismer + adapter.

Other question: The author mentioned that Prismer used RoBERTa as the decoder. By default, RoBERTa is a birectional transformer model. Did you revise RoBERTa to be a unidirectional model with casual masking?

---

> ### Author Response · Authors · 2023-11-21
> **Response 1/2**
>
> We appreciate the reviewer's constructive comments and highlight our contribution. Here, we resolve each of your concerns below.
>
> >The author has not mentioned some important related works, like Unified-IO (https://arxiv.org/pdf/2206.08916.pdf). Unified-IO also casts a bunch of vision-language tasks as sequence-to-sequence learning problems and trained a unified model that can solve all tasks. Different from Prismer, Unified-IO does not depend on the output from pretrained experts. Thus, the author should also compare with Unified-IO.
>
> Thanks for highlighting this important baseline. Originally, Unified-IO was not included in our study because we believed Unifed-IO primarily **focused on multi-task multi-modal learning** (also indicated by the reviewer), **instead of integrating multiple vision experts to enhance reasoning** (the primary focus of Prismer and other related works such as Visual Programming and ViperGPT). However, we acknowledge that Unified-IO presents an interesting and distinct approach to unifying vision experts, and we plan to discuss and incorporate Unified-IO in the final version of this paper.
>
> It's noteworthy that despite Unified-IO being pre-trained with **95 datasets for 22 tasks and having a 3x larger model parameter space** (its largest XL model), Prismer outperformed it in fine-tuned performance on VQAv2 and COCO caption tasks with a significant margin. Specifically, **Prismer achieved 78.4 in VQAv2 Accuracy compared to Unified-IO's 74.5; and 136.5 in COCO CiDER score compared to Unified-IO's 122.3**. This underscores Prismer's efficient and improved design strategy in incorporating vision experts.
>
>
> >The author indicated in the limitation section that Prismer is not good at adapting existing experts to new experts. This shows that performance of Prismer is tied to the choice of the experts, which impairs Prismer's flexibility.
>
> Yes, we are committed to providing an honest assessment of the limitations in the design of Prismer to guide future research directions. While incorporating additional experts may pose challenges, the flexibility of Prismer lies in its ability to **accommodate noisy experts or a reduced number of experts**. This flexibility was a key consideration in the original Prismer design, where we included six experts to cover essential and most commonly used vision tasks within the community.

---

> ### Author Response · Authors · 2023-11-21
> **Response 2/2**
>
> >As shown in the leftmost part of Figure 5, The accuracy boost obtained by adding more experts is diminishing. Prismer with 6 experts is 0.03 better than Prismer with 4 experts. Moreover, the right part of Figure 4 shows that the performance of Prismer falls significantly behind Flamingo and GIT.
>
> In Fig. 5(a), as mentioned in Section 5.1, we emphasise the observation that Prismer can easily allow for the addition of numerous experts (with constant memory), resulting in **enhanced performance without any performance degradation**. However, it's important to note that beyond a certain number of experts, the performance improvement plateaus, as **additional experts do not contribute additional useful information**.
>
> In Fig. 4 (right), our intention is to underscore that **Prismer's performance is inherently bounded by the size and quality of the pre-trained vision backbone**. A notable example is the switch from ViT-B to ViT-L, which yields a substantial absolute performance improvement of over 10\%. While we acknowledge that our performance is surpassed by Flamingo and GIT, considering **they are pre-trained with nearly three orders of magnitude more data**, the positive impact of switching the vision backbone suggests that Prismer's performance has the potential for **further improvement with a better vision backbone and increased pre-training data.**
>
>
> >There are not enough ablation study that justifies the design of the adapter. For example, will the model perform worse or better with full-finetuning? In fact, the author mentioned that Prismer is trained on significantly less amount of data than Flamingo / GIT. It is known that full fine-tuning works better than PEFT if we have a large number of data samples. The author can thus compare the performance difference between Prismer + full finetuning and Prismer + adapter.
>
> We have included **a thorough ablation study in Appendix D**, where we comprehensively evaluate all our architectural design choices and pre-training/fine-tuning strategies. The results confirm that our current fine-tuning strategies yield the best performance, aligning with observations in LiT (Zhai et al 2021). We will further highlight this in the main paper.
>
>
> >Other question: The author mentioned that Prismer used RoBERTa as the decoder. By default, RoBERTa is a birectional transformer model. Did you revise RoBERTa to be a unidirectional model with casual masking?
>
> Yes, that's exactly correct. Our empirical findings indicate that **RoBERTa outperforms OPT and Bloom in similar model sizes**. This observation is consistent with the findings in "Zero-Shot Video Question Answering via Frozen Bidirectional Language Models," where the use of DeBERTa was found to be superior to GPT-Neo and GPT-J. It's worth noting that with the emergence of more powerful auto-regressive models in recent months, such as Mistral-7B and Llama-1/2, this observation may no longer hold true.
>
> ---------------------
> Request Changes:
> 1. Added multi-task VLMs in the related work.
> 2. All alternative model designs and training/fine-tuning strategies have been already ablated in the paper (Appendix D).

---

> ### Author Response · Authors · 2023-12-18
> **Paper Update**
>
> Dear Reviewer,
>
> We have thoroughly revised the paper, addressing the comments and concerns you raised. A detailed response to your concerns and comments is provided above. We would appreciate your feedback on whether these updates sufficiently address your comments.
>
> Thank you!
> Authors

---

> > ### Comment · Reviewer_8Dhj · 2023-12-20
> >
> > Thanks for the response. My concerns have been addressed.

---

### Review · Reviewer_adRT · 2023-12-13

**Summary Of Contributions:**

The paper presents Prismer, a multimodal Transfromer-basedmodel  (vision-encoder + text-decoder). Both the vision and the text backbones use existing pre-trained models (ViT and RoBERTa, respectively), which are mostly frozen (i.e. not trained further) and that have been augmented with adaptors. The vision encoder also uses inputs from many vision expert models (e.g. depth estimation, object detection, ...; which are also frozen), and adds a new component, the "Experts resampler", that is used to post-process the outputs from the expert models.

The model is trained using a single image captioning loss on a mixture of five datasets: COCO, Visual Genome, Conceptual Captions, SBU captions and Conceptual 12M. It is later evaluated on multiple tasks: image captioning after fine-tuning, zero-shot captioning, and few-shot image classification. The reported results are especially good on Zer-shot image captioning, when compared against existing baselines.

**Audience:**

Yes

**Broader Impact Concerns:**

No concerns.

**Claims And Evidence:**

No

**Requested Changes:**

- Please, provide inference time information (e.g. #FLOPS/image, or ms/image on a given hardware) on the Zero-shot table in Figure 4. This is fundamental to put in context (in my opinion) the strongest results from the paper. Likewise, the few-shot results must also be put in context of the inference time, for the same reason (suggestion: provide a plot for 1shot or 5shot with FLOP/image or ms/image in the x-axis, and leave the rest of n-shots for the appendix). This is critical.

- Either remove the "pre-train pairs" column from Table 2, or add another column with "pre-train pairs" including the number of pre-training pairs used for the CLIP and RoBERTa pre-training. Otherwise, the comparison between methods using existing pre-trained models and approaches that train from scratch are strongly biased. Even better, as stated many times above, try to provide a comparison of inference runtime. This would strengthen the work significantly and make it more transparent.

**Strengths And Weaknesses:**

**Strengths**
- The paper includes quite a thorough ablation study of the different components introduced. Some of this analysis is presented in Section 5, but a lot more details are available in Appendix D. The ablation results have been limited to the BASE model, so it's unclear if some of the results carry on for bigger backbones, but this is understandable, given the cost of running such ablation experiments.

- The paper is well written, easy to follow, and with significant details to replicate the model and training settings. The authors also said to release the code and pre-trained models, which makes the adoption of the work and potential future follow-ups by other researchers much easier.

**Weaknesses**
- The technical novelty of the paper is quite limited. The only changes introduced to publicly available pre-trained models are the use of adapters (which are already quite popular and well known by the community), and the expert resampler (which is essentially the same as used by Perceiver & Flamingo, as the authors mention).

- The statement in the abstract: "Prismer achieves fine-tuned and few-shot learning performance which is competitive with
current state-of-the-arts, whilst requiring up to two orders of magnitude less training data" is not correctly demonstrated. Prismer uses pre-trained backbone models: CLIP and RoBERTa. The number of training examples used for these doesn't seem to be taken into account when comparing with state-of-the-art methods that train from scratch. See also the comment below, related to training costs depicted in Table 7.

- The fine-tuning results in Table 2 are not very solid. For the BASE size, when data is available BLIP is very often better or very close to the proposed method. For LARGE size, a similar thing happens with LEMON and BLIP. Overall, the benefits of Prismer in front of these other methods is not that clear.

- Fewshot results in the right of Figure 4 can be misleading: Prismer BASE has a total of 940M params, while ViT-B has about 100M params. Although the param count itself is not a perfect proxy, this number correlates well in most of cases with FLOPS/image and ms/image. This needs to be addressed (see requested changes).

- The comparison of training cost in Table 7 (appendix C) can be also a bit misleading, since Prismer uses already pre-trained models in its backbone, and that cost doesn't seem to be factored in, neither in the # of image-text pairs, nor in the # of PetaFLOP x days. For systems (like Prismer) using existing pre-trained models a comparison of inference runtime is a must.

- When adding more experts, we are also adding 1) more parameters and 2) more time complexity. In the ablation studying the effect of adding experts (Section 5.1 and 5.2), a better baseline would be to use a backbone encoder-decoder model with a comparable number of parameters and runtime. Otherwise, it's not clear whether the benefit comes from the "expertise" of the new components being added or just using more compute/parameters per image.

- Overall, it's not clear at all what the take-home-message is, for other researchers to build on top of the work.

---

> ### Author Response · Authors · 2023-12-18
> **Response 1/3**
>
> We appreciate the reviewer's detailed comments and highlight the clarity and reproducibility of our work. Here, we resolve each of your concerns below.
>
> >The technical novelty of the paper is quite limited. The only changes introduced to publicly available pre-trained models are the use of adapters (which are already quite popular and well known by the community), and the expert resampler (which is essentially the same as used by Perceiver & Flamingo, as the authors mention).
>
> While we acknowledge that each architectural component may not be deemed individually novel, it is crucial to emphasise that the overall design of Prismer demands significant engineering efforts to attain optimal memory efficiency. A comprehensive exploration of alternative designs employed in our experiments is detailed in Appendix D. Furthermore, we propose that the primary contribution of this work lies not in the novelty of individual architectural designs but rather in addressing the questions of **'how to scale down multi-modal learning' and 'how to better utilise existing pre-trained expert models?'**
>
>
> >The statement in the abstract: "Prismer achieves fine-tuned and few-shot learning performance which is competitive with current state-of-the-arts, whilst requiring up to two orders of magnitude less training data" is not correctly demonstrated. Prismer uses pre-trained backbone models: CLIP and RoBERTa. The number of training examples used for these doesn't seem to be taken into account when comparing with state-of-the-art methods that train from scratch. See also the comment below, related to training costs depicted in Table 7.
>
> We apologise for any confusion arising from our definition of pre-training cost. In our context, we define the number of pre-training data as the **data exclusively used during the creation of Prismer**. This excludes the original pre-training data used for training each individual expert model, **aligning with the same definition commonly used within the VLM community**. Notably, other state-of-the-art VLMs such as Flamingo, GIT-1/2, and BLIP-1/2 **also apply pre-trained vision/text models without considering these pre-training data as part of the training data**. This underscores the validity of our definition.
>
> We acknowledge the need to explicitly highlight and have clarified this definition in the paper to prevent any potential confusion.
>
> >The fine-tuning results in Table 2 are not very solid. For the BASE size, when data is available BLIP is very often better or very close to the proposed method. For LARGE size, a similar thing happens with LEMON and BLIP. Overall, the benefits of Prismer in front of these other methods is not that clear.
>
> We agree that BLIP and LEMON exhibit similar performance to Prismer. However, this observation precisely **underscores the effectiveness of Prismer's data efficiency**. By leveraging expert models, Prismer **achieves comparable performance to BLIP and LEMON while requiring only a fraction (x10/20 less) of the number of pre-training data**. This highlights Prismer's ability to achieve a performance level akin to existing models with significantly reduced pre-training data requirements.
>
> >Fewshot results in the right of Figure 4 can be misleading: Prismer BASE has a total of 940M params, while ViT-B has about 100M params. Although the param count itself is not a perfect proxy, this number correlates well in most of cases with FLOPS/image and ms/image. This needs to be addressed (see requested changes).
>
> We acknowledge the potential for confusion regarding the comparison. It's important to clarify that Prismer utilises the same ViT architecture for its vision encoder, ensuring an **identical representation space** for the few-shot image classification task when considering the rest of the model is frozen. Please note that the reported 980M parameter count for Prismer BASE includes 650M parameters for expert models, which are **solely employed for generating expert labels (and can be easily parallelised)**, and a 150M language model, used exclusively as **a classifier through a fixed prompt template**. While we recognise that scaling laws typically lead to improved performance with increased model parameter count, it's essential to highlight that in this case, the majority of the parameter count comes from expert models. We have further clarified this aspect in the updated version of the paper.

---

> > ### Author Response · Authors · 2023-12-18
> > **Response 2/3**
> >
> > >The comparison of training cost in Table 7 (appendix C) can be also a bit misleading, since Prismer uses already pre-trained models in its backbone, and that cost doesn't seem to be factored in, neither in the # of image-text pairs, nor in the # of PetaFLOP x days. For systems (like Prismer) using existing pre-trained models a comparison of inference runtime is a must.
> >
> > We understand the persisting concern, and we would like to reiterate that our definition of pre-training cost aligns with the practices within the VLM community. It refers specifically to the pre-training cost incurred exclusively in constructing our Prismer model. It is important to note that **among the 6 other state-of-the-art VLMs we compared with, only SimVLM is trained from scratch**, necessitating a substantial amount of pre-training data up to a billion scale. Conversely, **all other VLMs incorporate components that rely on existing pre-trained domain-specific backbones, whether in vision, text, or both.**
> >
> > These backbone models can sometimes **depend on proprietary data or may not be thoroughly explained in the paper (as seen in cases like PaLI and Flamingo)**. Consequently, comparing these models using pre-training costs based on the reviewer's definition becomes challenging. It's crucial to emphasise that our pre-training cost, measured in terms of # PetaFLOP days, **includes consideration for our 6 expert models.** We agree on the necessity to further clarify and highlight this definition in the main paper.
> >
> >
> > >When adding more experts, we are also adding 1) more parameters and 2) more time complexity. In the ablation studying the effect of adding experts (Section 5.1 and 5.2), a better baseline would be to use a backbone encoder-decoder model with a comparable number of parameters and runtime. Otherwise, it's not clear whether the benefit comes from the "expertise" of the new components being added or just using more compute/parameters per image.
> >
> > We appreciate the reviewer's intriguing perspective on this matter. However, we would like to assert that all 6 experts, as demonstrated in Appendix A, are exceptionally lightweight and can be **efficiently parallelised for inference (as implemented in our experiments).** Consequently, **having a larger model with the same parameter size as the combined parameter of these six expert models would not incur the same inference cost as the current design.**
> >
> > Acknowledging the reviewer's suggested changes, we agree with the importance of incorporating inference cost considerations into this discussion, as reflected in the updates made in our current version of the paper.
> >
> > >Overall, it's not clear at all what the take-home-message is, for other researchers to build on top of the work.
> >
> > We argue that Prismer stands out among VLMs due to its unique design strategy, which focuses on unifying existing expert models into a singular architecture, presenting a distinctive approach to architectural designs. In contrast, other approaches, exemplified by ViperGPT and Visual Programming, pivot towards converting multi-modal reasoning into sequential code generation within expert models accessed through APIs. This alternative direction offers a **zero-shot and highly interpretable solution** providing clarity on which expert contributes to a wrong prediction and consequently leads to an incorrect final answer. However, it comes with a trade-off of **slightly lower performance**. Prismer, in contrast, offers a **fine-tuned and less interpretable solution**, designed for robustness to expert predictions and **achieving higher performance.**
> >
> > We have meticulously examined and studied its performance across different expert models and are committed to releasing all inference and training code. **We believe that sharing our code will immensely benefit the community, enabling other users and researchers to easily reproduce results, explore variations, and access core model implementation details for advancing future VLM designs.**

---

> > > ### Author Response · Authors · 2023-12-18
> > > **Response 3/3**
> > >
> > > Requested Changes:
> > >
> > > We appreciate the reviewer's main concerns, which primarily revolve around the definition of pre-training and inference costs. In response, we have implemented the following changes:
> > >
> > > 1. Regrettably, we are unable to provide #Flops/image or ms/image in Fig. 4 due to **unclear model details and the non-public availability** of Flamingo and GIT models. In our comparison, we relied on their published results for few-shot ImageNet to facilitate meaningful comparisons.
> > >
> > > 2. We have **included our estimated inference cost** as an additional column in Table 7, alongside all other compared models. However, we will maintain **the original definition of pre-training cost** in our paper, excluding additional data required for each expert model. We have explained the reasons for this choice in previous comments. Importantly, even if we were to modify this definition, it is worth noting that **other VLMs would still exhibit larger pre-training costs due to the use of larger backbone architectures.** It's essential to emphasise that pre-training cost holds greater significance, as continuous advancements in the field allow for ongoing improvements in inference costs, such as through the implementation of low-bit quantisation techniques, which are active research areas in the current LLM community.
> > >
> > > 3. We have provided clarifications and highlighted the definition of our pre-training cost in the updated version of our paper.

---

### Author Response · Authors · 2023-12-18
**General Comments**

Dear Reviewers:

We express our sincere appreciation to all reviewers for their valuable comments and insightful suggestions that have contributed to the improvement of our paper. In response to the comments and requested changes, we have implemented the following updates (highlighted in colour orange):

1. As suggested by Reviewer 8Dhj, we have expanded our related work section to include multi-task VLMs, providing a more comprehensive overview of the literature in this domain.
2.  As suggested by Reviewer cr8h, we have added explanations regarding the selection of the 6 task-experts.
3. Addressing concerns raised by Reviewers adRT and cr8h, we have highlighted in the paper that the incorporation of expert models has a minimal impact on both inference and training costs. This clarification aims to address concerns and provide a clearer understanding of the computational efficiency of our approach.

4. Addressing concerns raised by Reviewer adRT, we have emphasised in the paper that the same representation space is employed in our few-shot ImageNet experiments.

5. As suggested by Reviewers adRT and cr8h, we have included explicit information on inference costs and provided further clarification on the definition of training costs in the updated paper.

We believe that these updates effectively address the concerns raised by the reviewers. We eagerly await further feedback and confirmation on whether the current version meets the expectations and requirements outlined by the reviewers.

---

### Decision · Action_Editor_X4hV · 2024-01-12

**Recommendation:** Accept as is

**Comment:**

In addition to the above, Reviewer cr8h's main reservation concerns novelty: "While the paper has some merits, it appears to be limited in terms of novelty." TMLR's [evaluation criteria](https://jmlr.org/tmlr/editorial-policies.html#:~:text=Evaluation%20criteria,the%20findings%20of%20this%20paper%3F) explicitly state that submissions should be accepted if they meet the "Claims and Evidence" and "Audience" criteria, even if the contributions or significance of the work is modest. All reviewers indicate in their official recommendation that the submission does satisfy the two criteria.

**Audience:**

The consensus among reviewers is that the submission is of interest to TMLR's audience:

* Reviewer 8Dhj: "The idea of tackling vision-language reasoning by ensembling models trained on different tasks is interesting."
* Reviewer adRT: "[...] when putting everything on the balance, I still find that the paper may be of value for a wide range of TMLR readers."

**Claims And Evidence:**

Reviewer adRT remains concerned that "some comparisons are done without properly taking into account pre-training cost", however they acknowledge that the submission's comparison approach is consistent with many works in the literature and finds that "the claims [made] are supported by the evidence (of course, taking into account the definition of "pre-training cost" that the authors make and after their clarifications)".

Reviewer 8Dhj notes that "[a]lthough scaling up the individual experts may further improve the performance, it is unclear if Prismer can outperform Flamingo / GIT given the current data points", but this is more of a comment on significance than on the empirical support of the claims made in the submission.